# Relation between Droplet Size Distributions and Physical Stability for Zein Microfluidized Emulsions

**DOI:** 10.3390/polym14112195

**Published:** 2022-05-28

**Authors:** Jenifer Santos, Luis Alfonso Trujillo-Cayado, Francisco Carrillo, María Luisa López-Castejón, María Carmen Alfaro-Rodríguez

**Affiliations:** 1Departamento de Ingeniería Química, Escuela Politécnica Superior, Universidad de Sevilla, c/Virgen de África, 7, E41011 Sevilla, Spain; cfuente@us.es (F.C.); llcastejon@us.es (M.L.L.-C.); 2Departamento de Ingeniería Química, Facultad de Química, Universidad de Sevilla, c/Profesor García González, 1, E41012 Sevilla, Spain; alfaro@us.es

**Keywords:** emulsion, droplet size distribution, microfluidization, Pickering emulsion, response surface methodology, Turbiscan Stability Index, zein

## Abstract

Zein, a subproduct of the food industry and a protein, possesses limited applications due to its high hydrophobic character. The objective of this research was to investigate the influence of homogenization pressure and cycles on the volumetric mean diameter (D_4,3_), span values, and Turbiscan Stability Index (TSI) using the response surface methodology for microfluidized emulsions containing zein as a unique stabilizer. Results showed that homogenization pressure seems to be the most influential parameter to obtain enhanced physical stability and droplet size distributions, with the optimum being 20,000 psi. Interestingly, the optimum number of cycles for volumetric diameter, span value, and TSI is not the same. Although a decrease of D_4,3_ with number of cycles is observed (optimum three cycles), this provokes an increase of span values (optimum one cycle) due to the recoalescence effect. Since physical stability is influenced by D_4,3_ and span, the minimum for TSI is observed at the middle level of the cycles (2 cycles). This work highlights that not only volumetric diameter, but also span value must be taken into consideration in order to obtain stable zein emulsions. In addition, this study wants to extend the limited knowledge about zein-based emulsions processed with a Microfluidizer device.

## 1. Introduction

Emulsions are biphasic systems, where droplets are dispersed into a continuous medium. These systems are thermodynamically unstable. However, they can be kinetically stable. In emulsions, there are different destabilization processes, such as flocculation, creaming, sedimentation, coalescence, Ostwald ripening, and phase inversion. Droplet size distribution is an emulsion property that can determine its physical stability since it influences rheology, creaming rate, and microstructure. Interestingly, droplet size distribution can be quantified using different parameters, e.g., Sauter diameter, volumetric diameter, span, or uniformity. There are several studies that relate these parameters with physical stability and rheology [1,2,3]. Normally, smaller medium droplet sizes and narrower droplet size distributions provokes higher viscosities and enhanced physical stability.

Zein protein, a food by-product obtained from corn, has attracted much attention due to its multiple applications [4]. Among these, it is important to highlight the development of zein-based products as drug delivery or encapsulation systems [5,6]. However, zein has limited application for stabilizing Pickering emulsions due to its high hydrophobic character [7,8]. In addition, zein is only soluble in highly alkaline (pH > 11) solutions [9] and ethanol. Hence, Pickering emulsions stabilized by zein particles alone are scarcely efficient and they could cream easily. Several attempts have been made to solve this problem, e.g., by the incorporation of polysaccharides or combination with other proteins to modify the zein-based composite particles, aiming to enhance the physical stability and properties of zein-based Pickering emulsions [10,11]. Interestingly, zein shows a helical wheel quaternary structure [12]. Different treatments (physical, chemical, and enzymatic) are used to modify the structure and conformation of proteins, which can modify the physicochemical and functional properties [13].

Different mechanical devices have been used in order to develop emulsions with an appropriate droplet size distribution: rotor-stators, microfluidizers, ultrasonics, or high-pressure valve homogenizers [14]. The base of these devices is the use or an external energy to break large droplets into smaller ones during emulsification. Microfluidization is specialised for the formation of uniform emulsions or nanoemulsions. This device uses a high-pressure pump that forces the dispersed and continuous phases through an interaction chamber composed by small channels named microchannels. An impingement area along these microchannels produces fine droplets, normally smaller than one micron [15]. In addition, microfluidization can be used as a physical modification method for proteins and dietary fibers. In this way, ternary and quaternary structures of the proteins can be modified as a result of high shear forces, resulting in an improvement of functional properties of proteins [16]. Microfluidization has been used to enhance functional properties of various food products, such as hazelnut skin fiber [17], high methoxyl pectin [18], and whey protein [19]. The study of microfluidization or ultrasonication applied to the development of zein-based emulsions is very limited [20,21].

The response surface methodology (RSM) was first reported by Box and Wilson (1951) [21]. Nowadays, this methodology is used in multiple fields, such as microbiology [22], chemical engineering [23], or environmental sciences [24]. The base of RSM is to use some designed experiments to explore the relations between independent variables (X) and the response variable (Y) and obtain an optimum using linear or second-degree models. This statistical technique models and analyses problems in which a response of interest (Y) is influenced by other variables (X_1_, X_2_, …). 

The main objective of this work was to evaluate how the span parameter and volumetric diameter of zein-based emulsions can influence the physical stability of emulsions using the response surface methodology. The minimum of volumetric diameter and span parameter were determined and related to the physical stability measured by Turbiscan Stability Index (TSI). This study wants to extend the limited knowledge about concentrated emulsions stabilized only with zein and developed by microfluidization. It also contributes to the study of microfluidized food grade emulsions in connection with their droplet size distributions and physical stability, which is of paramount importance for their handling properties.

## 2. Materials and Methods

### 2.1. Materials

Zein protein was provided by Sigma Aldrich (St. Louis, MO, USA). Sunflower oil containing 40 wt.% of oleic acid was obtained from Coreysa company (Sevilla, Spain). All emulsions were prepared using deionized water. 

### 2.2. Functional Properties Determination: Solubility and Zeta Potential

Suspensions of 1 g zein/100 mL were prepared by adding the protein to water adjusted to various pH values ranging from 1 to 13 with 0.5 M HCl or 0.5 M NaOH. Afterwards, zein solubility was quantified as described by Peterson [25]. For the determination of the Z-potential, a Zetasizer Nano ZS (Malvern Instruments, Malvern, UK) was used, using the Smoluchowski equation [26]. The measures were carried out in triplicate.

### 2.3. Microfluidization of Food Emulsions Formulated with Zein Protein

The formulation used for the preparation of emulsions was 0.5 wt.% of zein, 50 wt.% of sunflower oil and deionized water. Firstly, the continuous phase was prepared by dispersing zein protein into deionized water and adjusting the pH to 11.5 [9]. Then, a coarse emulsion (batches of 250 g) was prepared, at room temperature, using a Silverson L5M (Silverson, Chesham, UK) for 90 s at 8000 rpm. Finally, finer emulsions were homogenized using a Microfluidizer M110P (Microfluidics company, Westwood, MA, USA) at different processing parameters (Table 1). The microfluidizer device was used with a configuration of Y + Z and a refrigeration temperature of 20 °C.

### 2.4. Design of Experiments

An experimental design and response surface methodology were used to analyze the relationship between the dependent variables (volumetric mean diameter, span, and Turbiscan Stability Index) and independent variables (number of cycles and homogenization pressure). The experimental design consisted of three levels and two factors, generating 32 experiments. This results in 9 experiments and 2 additional replicates of the central point (see Table 1). Every experiment was conducted by duplicate. All the data were analyzed with a one-way analysis of variance (ANOVA) at a 95% confidence level. All the experimental design and data analyses were performed using the Echip software (Experimentation by Design, Wilmington, DE, USA).

### 2.5. Laser Diffraction Measurements 

In order to characterise droplet size distribution of the emulsions developed, a Malvern Mastersizer 2000 (Malvern, UK) was used. Furthermore, volumetric diameter (D_4,3_) was used in order to quantify and compare the mean diameters of the emulsions developed. Finally, span parameter values were used to quantify the polydispersity of the droplets created.
(1)$$D_{4,3}=\overset{N}{\underset{i=1}{\sum }}n_{i}d_{i}^{4}/\overset{N}{\underset{i=1}{\sum }}n_{i}d_{i}^{3}$$
(2)$$\mathrm{span}=\frac{D_{90}-{\;D}_{10}}{D_{50}}$$
where d_i_ is the droplet diameter, N is the total number of droplets, ni is the number of droplets having a diameter di, and d_90_, d_50_, d_10_ are the diameters at 90%, 50%, and 10% cumulative volume. 

### 2.6. Multiple Light Scattering Technique

Turbiscan Stability Index (TSI) is a parameter that allows physical instability to be quantified and compared. It has been calculated following Equation (3):(3)$$\mathrm{TSI}=\underset{j}{\sum }\left|{\mathrm{scan}}_{\mathrm{ref}\;}\left(h_{j}\right)-{\;\mathrm{scan}}_{i}\left(h_{j}\right)\right|$$
where scan_ref_ and scan_i_ are the initial backscattering value and the backscattering value at a specific time, respectively, and h_j_ is a specific height in the measuring cell. 

In order to obtain this parameter, backscattering measurements were carried out for the samples developed at different aging times using Turbiscan Lab Expert (Formulaction, France). 

## 3. Results and Discussion

Figure 1 shows the solubility of zein in water and the Z-potential as a function of pH. On the one hand, zein is in a low solubility condition throughout the investigated pH range from 2.5 to 7. In addition, a significant increase for solubility was found at pH values higher than 8. At these pH values, zein side chains contain more negative net charge, influencing its structural stability, i.e., giving rise to higher solubility. This behavior could be explained by the electrostatic interactions produced by the side chains with ionizing properties as well as hydrogen bond formation with the solvent. On the other hand, the potential value Z follows a downward trend as the pH increases, starting the highest value coinciding with pH 2.5 (lowest value evaluated) and reaching its minimum at a pH of 11.5. The most significant zeta potential drop is between pH 8 and pH 9.5. From this, it can be deduced that the isoelectric point of the protein will be in this range. This value coincides with what was found by other authors in the literature who establish that the isoelectric point is between 5 and 9 [27]. However, increasing the pH to 11.5 results in an improved solubility. Hence, pH 11.5 is selected for the evaluation of emulsion development. High ionic strength decreased the solubility and emulsifying activity of zein suspensions [28].

Figure 2 shows the droplet size distribution for emulsions containing 50 wt.% sunflower oil and 0.5 wt.% of zein (ration 1:10) as a function of microfluidization parameters studied (pressure and cycles). First of all, the droplet size distribution of the pre-emulsion is bimodal, while microfluidized emulsions show a monomodal distribution. Secondly, the use of microfluidization provoked a clear reduction of droplet sizes, thus proving its importance. Furthermore, there is a decrease of droplet size when pressure increases from 5000 to 15,000 psi after just one cycle. However, an increase of droplet size is observed above 15,000 psi. This fact is a clear sign of recoalescence due to over-processing. This usually occurs using microfluidization in emulsions where the protein/surfactant does not entirely cover the oil-water interface [29]. Hence, this result suggests that there is a lack of protein in the interface. In order to obtain a deeper insight into these results, Figure 3 is illustrated.

Variation of the volumetric diameter (D_4,3_) with homogenization pressure for different numbers of cycles is shown in Figure 3. The clear decrease in droplet size from 5000 to 15,000 psi is also observed here. In addition, a small increase in droplet size highlights the slight recoalescence from 15,000 to 25,000 psi. Whereas a big reduction of droplet size is observed from one to two cycles at 5000 psi. This is not noticed at 15,000 or 25,000 psi, and could be due to the lack of protein concentration to stabilize smaller droplets.

Results obtained for volumetric diameter and span values as a function of homogenization pressure (P) and cycles (C) are shown in Table 1. This table also illustrates the experiments design that has been carried out. These results have been modelled and optimized using the response surface methodology (RSM). On the one hand, the relation between volumetric diameter with microfluidized parameters is indicated in Figure 4 and in Equation (4). Equation (4) states that volumetric diameter fits a quadratic function of P and C with a determination coefficient (R^2^) of 0.99. This coefficient suggests the great correlation between the experimental results and the model.
(4)$$D_{4,3}\left(\mu m\right)=1.22-2.92\cdot P-0.73\cdot C+0.55\cdot P\cdot C+3.57\cdot P^{2}$$

Furthermore, *F* critical (F_crit_), which determines the significance of the groups of variables, is higher than *F* lack of fit (F_lof_) with *p* = 0.05, which tests how well the model fits the data. This fact is a clear indication of the suitability of the model. Volumetric diameter was sensitive to the homogenization pressure and number of cycles. Analyzing the coefficients of Equation (4), homogenization pressure seems to be more influential than cycles for volumetric diameter. These trends are clearly seen in Figure 4. At intermediate pressures, and especially at high pressures, the influence of the number of cycles on the diameters is not very significant. However, regardless of the number of cycles, volumetric diameters vary significantly with homogenization pressure. Taking this model into account, the minimum of volumetric diameter was at 20,000 psi and three cycles.

On the other hand, the relation between span values obtained with homogenization pressure and number of cycles is indicated in Figure 5 and in Equation (5). This correlation is also a quadratic one with more influence of homogenization pressure. In this case, the value of R^2^ is 0.89, showing a good fit. This fit presents a minimum at 20,000 psi and one cycle. In addition, there is an increase of span values from one to three cycles at 15,000 and 25,000 psi (see Table 1). The latter could be due to the recoalescence effect abovementioned. The increase of number of cycles provoked a reduction of volumetric diameter but also an increase of span values. This fact has been reported for other emulsions containing proteins [30]. It is important to notice that physical stability of emulsions is not only influenced by mean diameters, but also by span values [31].
(5)$$\mathrm{span}=1.126-0.211\cdot P+0.137\cdot C+0.184\cdot P\cdot C+0.481\cdot P^{2}$$

Figure 6 shows the variation of backscattering (BS) with height of the measuring cell as a function of aging time for the pre-emulsion. There is a big drop in the lower part of the measuring cell, that is related to a clarification process. Hence, the droplets are moving to the upper part of the vial. This is the definition of the creaming process. This process occurs in 1 hour and a half in this pre-emulsion. Therefore, the pre-emulsion showed a very poor stability due to its wide droplet size distribution centered at a big droplet size. 

Figure 7 illustrates the variation of backscattering (ΔBS) profile with height of the measuring cell as a function of aging time for the emulsion processed at 15,000 psi and two cycles. A decrease of ΔBS with storage time in the bottom and upper part of the vial is observed. These facts are a clear indication of a clarification process in the bottom part and a higher concentration of droplets in the upper part, i.e., a creaming process. Two big drops are observed, pointing to the destabilization process by creaming. Compared to the pre-emulsion (Figure 6), it seems that emulsions processed using the microfluidizer showed an enhanced physical stability. 

Figure 8 illustrates the relation between TSI values with homogenization pressure and number of cycles. In addition, Equation (6) states that TSI values follows a quadratic function with a determination coefficient (R^2^) of 0.99. Furthermore, the homogenization pressure is also the most influential variable. The trend of the TSI with respect to homogenization pressure and the number of cycles observed in Figure 8 is similar to that shown in Figure 4 for volumetric diameter. Thus, while the TSI is heavily influenced by pressure regardless of the number of cycles, it does not vary significantly with the number of cycles at intermediate and high pressures. The minimum is observed at 20,000 psi and two cycles.
(6)$$\mathrm{TSI}=5.89-3.99\cdot P-0.45\cdot C+0.55\cdot P\cdot C+4.88\cdot P^{2}$$

Analyzing every minimum of the models, it is interesting to note that 20,000 psi is a key factor to obtain a good droplet size distribution and enhanced physical stability. However, while the minimum volumetric diameter is obtained at three cycles, the minimum span is obtained at one cycle and the minimum TSI value at two cycles. This fact is explained by the recoalescence that these emulsions present. There are more cycles in the microfluidizer and a lower volumetric diameter, but higher span values. Interestingly, physical stability is influenced by the two factors. In this way, the minimum for TSI is observed at the middle level of the cycles (two cycles). 

## 4. Conclusions

Recently, the interest in the use of zein in the development of emulsions and bio-based delivery systems has been increasing. In this study, a solubility study has proven that pH 11.5 is suitable to prepare emulsions with zein. Optimization of the microfluidization parameters (homogenization pressure and cycles) in order to minimize the volumetric diameter and the span value of concentrated emulsions containing zein and sunflower oil was carried out. It has been proven that microfluidization provokes monomodal distributions regardless of homogenization pressure and number of cycles, improving the droplet size distributions (DSD) of the pre-emulsion. In addition, overprocessing has been observed for emulsions which have been summited above 15,000 psi, suggesting that zein does not entirely cover the interface. The response surface methodology has proven its importance to obtain not only a clear minimum of volumetric diameter, span value, and TSI, but also the key processing parameter (homogenization pressure) that influences these properties. Interestingly, the minimum of volumetric diameter, span, and TSI are not the same. This fact points out that volumetric diameter and span have to be taken into consideration in order to obtain stable zein emulsions. This study has revealed the impact of microfluidization on concentrated emulsions formulated only with zein as stabilizer, highlighting the importance of the selected pressure to develop these systems. 

## Figures and Tables

**Figure 1 polymers-14-02195-f001:**
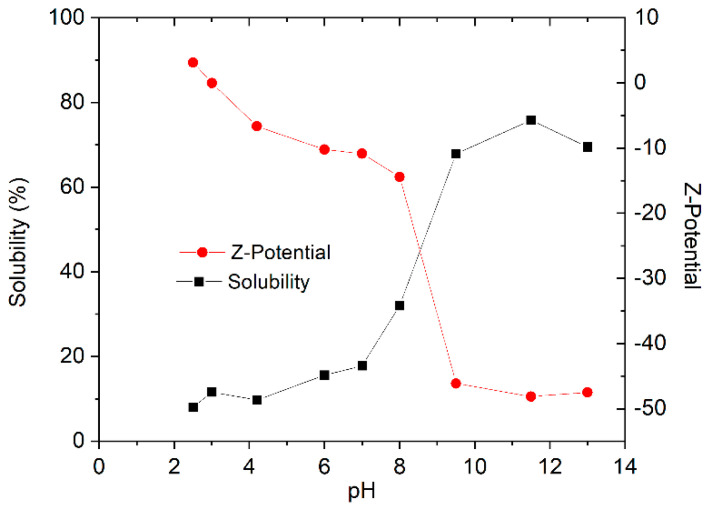
Zein solubility and zeta potential values as a function of pH.

**Figure 2 polymers-14-02195-f002:**
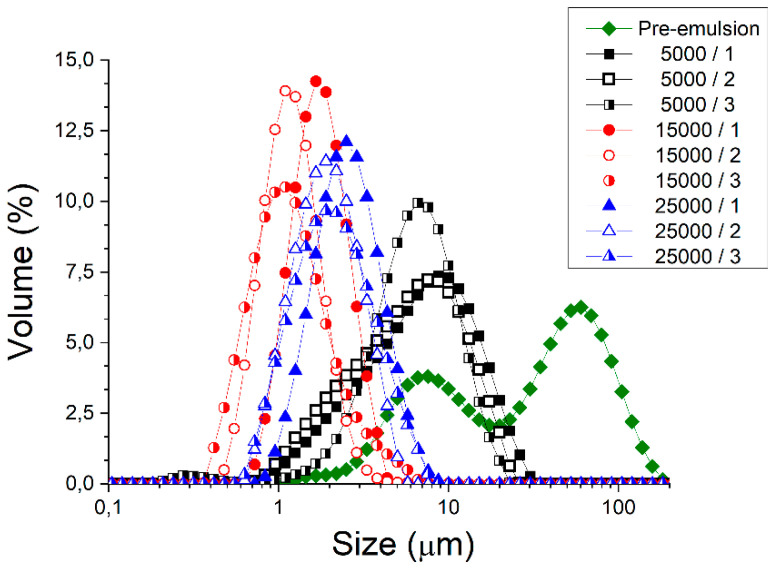
Droplet size distributions for emulsions developed formulated with zein as a function of homogenization pressure and passes in the microfluidizer.

**Figure 3 polymers-14-02195-f003:**
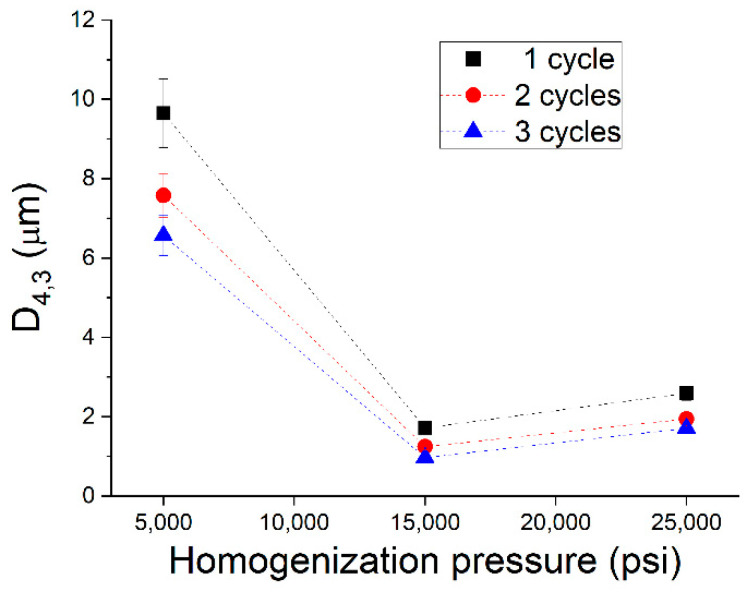
Volumetric mean diameters (D_4,3_) for emulsions developed formulated with zein as a function of homogenization pressure and number of cycles in the microfluidizer.

**Figure 4 polymers-14-02195-f004:**
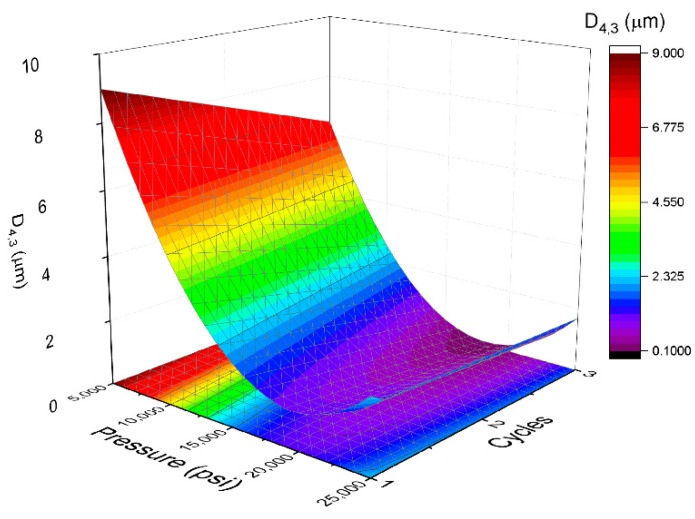
Response surface 3D plot of volumetric diameter as a function of pressure and cycles used in the microfluidizer.

**Figure 5 polymers-14-02195-f005:**
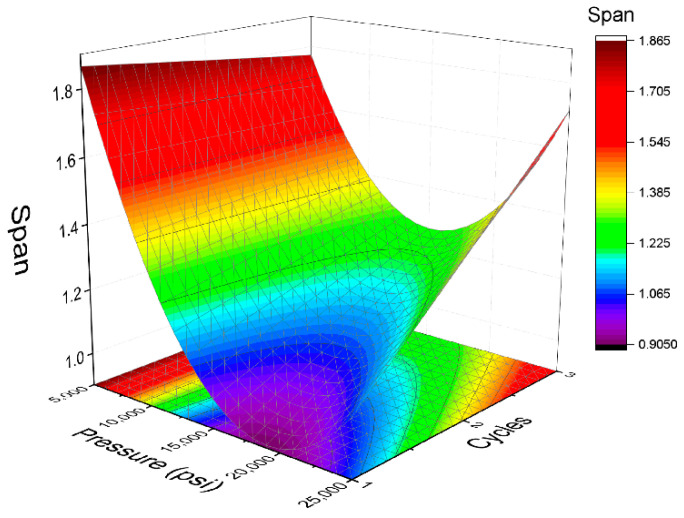
Response surface 3D plot of span values as a function of pressure and cycles used in the microfluidizer.

**Figure 6 polymers-14-02195-f006:**
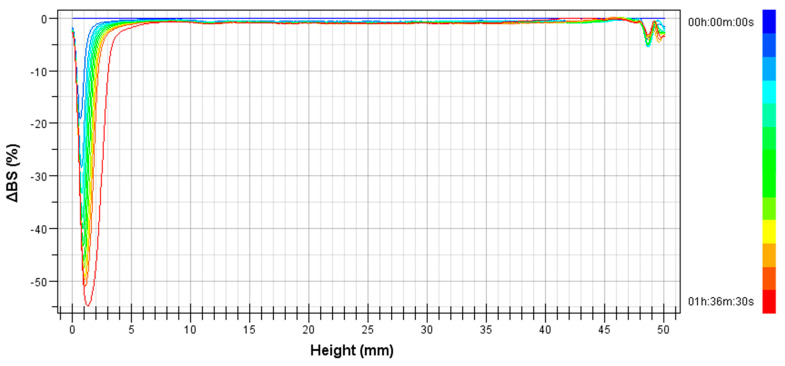
Variation of Backscattering (ΔBS) with height of the measuring cell as a function of aging time for the pre-emulsion.

**Figure 7 polymers-14-02195-f007:**
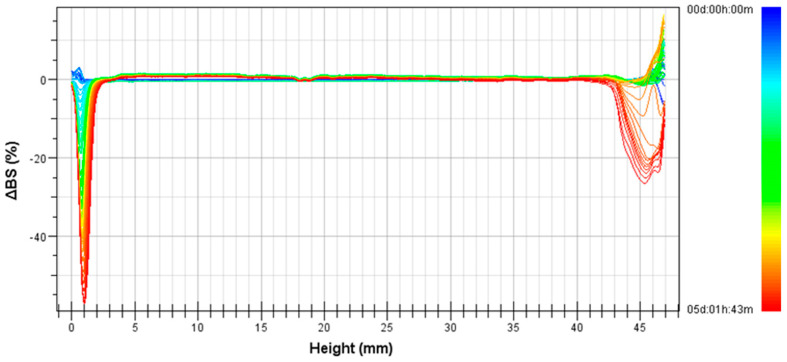
Variation of Backscattering (ΔBS) with height of the measuring cell as a function of aging time for the microfluidized emulsion processed at 15,000 psi and two cycles.

**Figure 8 polymers-14-02195-f008:**
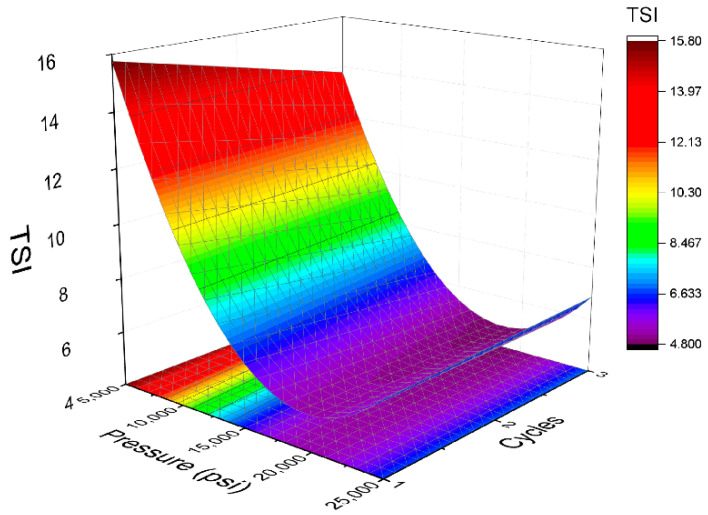
Response surface 3D plot of Turbiscan Stability Index (TSI) values as a function of pressure and cycles used in the microfluidizer.

**Table 1 polymers-14-02195-t001:** Experimental design, processing parameters, volumetric mean diameters (D_4,3_), span and Turbiscan Stability Index (TSI) values for all emulsions studied.

Sample	X_1_	X_2_	Pressure	Cycles	D_4,3_ (μm)	span	TSI
1	−1	−1	5000	1	9.65 ± 0.87	1.980	16.35
2	−1	0	5000	2	7.58 ± 0.55	1.925	13.51
3	−1	1	5000	3	6.57 ± 0.51	1.718	13.97
4	0	−1	15,000	1	1.72 ± 0.11	1.015	6.22
5	0	0	15,000	2	1.22 ± 0.07	1.051	6.02
6	0	0	15,000	2	1.25 ± 0.09	1.071	6.09
7	0	0	15,000	2	1.27 ± 0.08	1.06	6.05
8	0	1	15,000	3	0.97 ± 0.06	1.629	6.10
9	1	−1	25,000	1	2.59 ± 0.18	1.228	6.75
10	1	0	25,000	2	1.94 ± 0.16	1.455	6.58
11	1	1	25,000	3	1.71 ± 0.13	1.702	6.55

## Data Availability

Not applicable.

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
