# Peer review of "Relation between Droplet Size Distributions and Physical Stability for Zein Microfluidized Emulsions"

_polymers, 2022, doi:10.3390/polym14112195_

Round 1

Reviewer 1 Report

The corresponding manuscript has reported results in the field of application of zein in emulsification process by using microfluidization. The results are well presented, figures and tables are appropriate, but the scientific novelty of the results is rather low. Several similar studies are already available concerning this specific topic and the conclusions drawn. Therefore, after careful evaluation of the submitted manuscript, I suggest rejection due to the substantial lack of novelty. 

Author Response

We thank the reviewer for appreciating our work. Although there are several studies concerning zein-based emulsions, the knowledge about microfluidized emulsions stabilized only with zein and formulated with a high oil content is very limited. As suggested by other reviewer, we have improved the abstract and introduction of the manuscript to better highlight the novelty of the research:

  • Abstract:

In addition, this study wants to extend the limited knowledge about zein-based emulsions processed with a Microfluidizer device.

  • Introduction:

In addition, this study wants to extend the limited knowledge about concentrated emulsions stabilized only with zein and developed by microfluidization. This study wants to extend the limited knowledge about concentrated emulsions stabilized only with zein and developed by microfluidization. It also contributes to the study of microfluidized food grade emulsions in connection with their droplet size distributions and physical stability, which is of paramount importance for their handling properties.

Reviewer 2 Report

The manuscript presents means to study the influence of different parameters on physical stability of emulsions.

The manuscript can be published after minor revision, according to the list below:

  1. The authors should not use abbreviations unless they first specify what the abbreviations stand for. For example BS (r. 208), DSD (r. 252).
  2. Please specify for equation (2) what represents D90, D10 and D50. Also, what is the meaning of F­crit and Flof that you mentioned at r. 183.
  3. r. 149: “Furthermore, there is a decrease of droplet size when pressure increases from 5000 to 15000 psi…”
  4. r. 161: “Variation of the volumetric diameter (D4,3) with homogenization pressure for different number of cycles is shown in figure 3.”

Author Response

We sincerely thank the reviewer for the help in improving the manuscript.

  1. The meaning of the abbreviations has been included in the revised version of the manuscript.
  2. Done
  3. The sentence has been corrected as suggested.
  4. The sentence has been corrected as suggested.

Reviewer 3 Report

The manuscript entitled “Relation between droplet size distributions and physical stability for zein microfluidized emulsions” represents the effort to investigating the influence of homogenization pressure and cycles on volumetric mean diameter, span values and Turbiscan Stability Index (TSI) using response surface methodology for microfluidized emulsions containing zein as unique stabilizer. The authors should address the following points before further consideration for publication in polymers

  1. On page number 4, lines number 149, the authors discussed “However, decreasing the pH to 11.5 results in an improved…” the authors should change decreasing to increasing.  
  2. On page number 6 in Figure 4, the authors should change the graph or emphasize the relationship between number of cycles and volumetric diameter, there is no change from cycle 1 to 3. It would be easy to understand to the readers.
  3. On page number 9 in Figure 8 the authors should change the graph or emphasize the relationship between number of cycles and TSI, there is no change from cycle 1 to 3. It would be easy to understand to the readers.

Author Response

The manuscript entitled “Relation between droplet size distributions and physical stability for zein microfluidized emulsions” represents the effort to investigating the influence of homogenization pressure and cycles on volumetric mean diameter, span values and Turbiscan Stability Index (TSI) using response surface methodology for microfluidized emulsions containing zein as unique stabilizer. The authors should address the following points before further consideration for publication in polymers.

Thanks for the reviewer’s comment

  • On page number 4, lines number 149, the authors discussed “However, decreasing the pH to 11.5 results in an improved…” the authors should change decreasing to increasing.

Thanks for the observation. We have revised it.

  • On page number 6 in Figure 4, the authors should change the graph or emphasize the relationship between number of cycles and volumetric diameter, there is no change from cycle 1 to 3. It would be easy to understand to the readers.
  • On page number 9 in Figure 8 the authors should change the graph or emphasize the relationship between number of cycles and TSI, there is no change from cycle 1 to 3. It would be easy to understand to the readers.

Thanks for the reviewer’s suggestion. Figures that represent 3D response surface results are difficult to modify. However, we have included new sentences in the results discussion section in order to clarify this fact (Line 197 and Line 246).

“These trends are clearly seen in Figure 4. At intermediate pressures and especially at high pressures, the influence of the number of cycles on the diameters is not very significant. However, regardless of the number of cycles, volumetric diameters vary significantly with homogenization pressure.”

“The trend of the TSI with respect to homogenization pressure and the number of cycles observed in Figure 8 is similar to that shown in Figure 4 for volumetric diameter. Thus, while the TSI is heavily influenced by pressure regardless of the number of cycles, it does not vary significantly with the number of cycles at intermediate and high pressures.”

Reviewer 4 Report

The manuscript, relation between droplet size distributions and physical stability for zein microfluidized emulsions, describes the utilization of response surface methodology to evaluate the influence of span parameter and volumetric analysis on the physical stability of zein protein-based emulsion. There are a few points that need to be discussed before the publication of this manuscript.

  1. The author should emphasize the novelty of the method in the abstract and the last paragraph of the introduction.
  2. Figure 1, there is a sudden increase in the solubility at pH 8 as compared to pH 7. However, at these pHs, there is no significant difference in the surface charges of the emulsion (Z-potential result). There will be some OH- groups in the solution but further increase in the pH does not significantly increase the solubility. More discussion is needed that how the pH values influence the solubility.
  3. Only the influence of pH on solubility was observed. How other factors such as temperature and ionic strength of the aqueous medium will affect the solubility of zein protein and stability of the emulsion.
  4. Introduction, the discussion of different mechanical devices on page 2 needs references.

Author Response

We thank the reviewer for her/his constructive comments and suggestions on our manuscript.

  1. In order to empathize the novelty of this manuscript, a new sentence has been included in the abstract and a new paragraph in the introduction:

  • Abstract:

In addition, this study wants to extend the limited knowledge about zein-based emulsions processed with a Microfluidizer device.

  • Introduction:

In addition, this study wants to extend the limited knowledge about concentrated emulsions stabilized only with zein and developed by microfluidization. This study wants to extend the limited knowledge about concentrated emulsions stabilized only with zein and developed by microfluidization. It also contributes to the study of microfluidized food grade emulsions in connection with their droplet size distributions and physical stability, which is of paramount importance for their handling properties.

  1. Taking into account the reviewer´s comment, we have repeated solubility and zeta-potential measurements at pH = 8 because we think there could be an error or incongruity with the results obtained. As expected, the solubility value provided above was erroneous. That is why we have modified Figure 1. However, we agree with the referee that it's necessary to include more information and discussion about zein solubility. The following paragraph has been included in the revised manuscript as suggested:

At these pH values, zein side chains contain more negative net charge influencing its structural stability giving rise to higher solubility. This behavior could be explained by the electrostatic interactions produced by the side chains with ionizing properties as well as hydrogen bond formation with the solvent.

  1. The effect of temperature has been taken into account for a future study, not only on solubility but on emulsifying properties and emulsion stability. The influence of ionic strength has been reported by other authors. However, we have included a new sentence and a new reference:

High ionic strength decreased the solubility and emulsifying activity of zein suspensions.

Wu, Y. V. (2001). Emulsifying activity and emulsion stability of corn gluten meal. Journal of the Science of Food and Agriculture81(13), 1223-1227.

  1. A new reference has been including to clarify this fact:

Rayner, M., & Dejmek, P. (Eds.). (2015). Engineering aspects of food emulsification and homogenization. CRC Press.

Round 2

Reviewer 1 Report

The authors have not improved the manuscript in terms of novelty of the presented results, therefore, I suggest rejection of the submitted manuscript.

Author Response

The authors deeply regret that the reviewer does not consider the study to be novel. However, taking into account the suggestion of the publisher, we have responded again to the reviewers as well as a new reviewer. In any case, many thanks to the reviewer for his/her comments.
